# Robust Group PCA for Separable Noise: An Argument for Subject-Level PCA

Samuel Oriola*, Calvin McCurdy†, Bradley T. Baker§, Vince D. Calhoun‡§ and Rogers F. Silva§

*Department of Computer Science, †Department of Mathematics and Statistics
Georgia State University, Atlanta, Georgia, USA 30302
‡Department of Electrical and Computer Engineering
Georgia Institute of Technology, Atlanta, Georgia, USA 30332
§Tri-Institutional Center for Translational Research in Neuroimaging and Data Science (TReNDS)
Georgia State University, Georgia Institute of Technology, and Emory University, Atlanta, GA, USA 30303
{soriola1, cmccurdy5, bbaker43, vcalhoun, rsilva}@gsu.edu

*Abstract*—**Functional magnetic resonance imaging (fMRI) captures whole-brain function with high spatial resolution and has driven discoveries in brain connectivity across age, gender, mental illnesses, and developmental stages. As fMRI datasets grow in number, aggregation methods such as averaging or low-rank approximations are increasingly more likely to lose subject-specific details, potentially biasing group estimates and misrepresenting individuals, which in turn limits replication and reduces the translational utility of findings——especially among minorities. Group principal component analysis (PCA) is the *de facto* tool for aggregating datasets, with tools like FSL and GIFT supporting various implementations that have shaped neuroimaging studies for decades. Yet, the impact of subject variability on the group-level estimate, as well as the ensuing subject specificity, remain unquantified. This study aims to identify computational strategies that improve the accuracy and robustness of group-level representations. Three common group PCA implementations are considered: 1) simple concatenation, 2) concatenation with subject sum of squares normalization, and 3) concatenation with subject-level PCA whitening. Simulated scenarios test these methods to identify optimal approaches for group dimensionality reduction while preserving the ground-truth group mean information. Our results demonstrate that concatenation with subject-level PCA whitening achieved the best overall approximation of the ground-truth group mean, with performance differences largely driven by separable noise.**

*Index Terms*—**Group PCA, ICA, Data Analysis, Functional MRI, Multivariate**

## I. INTRODUCTION

Functional magnetic resonance imaging (fMRI) is a key healthcare technology that captures indirect measurements of neural signals in-vivo with unmatched spatial resolution. It has catalyzed discoveries in brain connectivity, for instance, unveiling default mode network interactions with other resting state networks [1] and dynamic dysconnectivity in schizophrenia [2]. With its increased adoption, large multi-dimensional fMRI datasets have become commonplace. These datasets are rich with subject-specific details, which must be optimally preserved in group-level analyses to provide the necessary insights for personalized treatment and long-term care options. Without the preservation of subject-specific information, group data can become biased and misrepresent individuals, hamper

The authors thank the NSF CREST D-MAP Program, NSF2112455.

interpretations, curtail the replication of findings, and perpetuate the exclusion of minorities in clinical research.

Group-level representations are notably hard to obtain due to a simple fact: all brains are different, just like faces. Despite marked progress in digital warping of individual images into standardized brain templates, the resulting "spatial maps" still vary significantly across subjects. Besides expected subject variation, factors such as image quality, resolution, noise, head movement, or even physiological interference from heartbeat and breathing, lead to registration and warping imperfections that can hamper accurate group-level aggregation.

Group principal component analysis (PCA) is a staple tool for group-level aggregation and is ubiquitous in neuroimaging. The FSL [3] and GIFT [4] toolboxes readily offer group PCA implementations and have supported downstream group-level studies in thousands of publications over the past two decades. Implementations of group PCA vary, however, and no study so far has pragmatically evaluated the impact of subject variability on the quality of group representations obtained with different implementations of group PCA (gPCA). This work explores the limits of gPCA, aiming to identify improvements for group-level representations and make recommendations for future research. Such improvements could enhance downstream analyses in neuroimaging toolboxes, increase the accuracy of group-level representations, enable more robust feature identification of spatial maps, and inspire novel insights into brain function that would be unattainable without preserving more of the underlying subject-specific details.

In this work we briefly review prior related work in Section II and describe our proposed approach and rationale in Section III. Section IV presents results and discusses the effects of different implementations of gPCA, while Section VI reviews our findings and outlines future investigations.

## II. PRIOR RELATED WORKS

### A. Preliminaries

Motivated by the hope that high variance might be synonym with useful information, the basic model for noiseless principal component analysis (PCA) of real-valued data [5] stems from finding the underlying direction $\mathbf{a}^\star$ along which the data

variance is maximal, which is also equivalent to minimizing the variance in the subspace orthogonal to $\mathbf{a}^\star$ [6], [7]. The former is formulated as:

$$\mathbf{a}^\star = \underset{\mathbf{a}}{\arg\max}\ \mathbf{a}^\top \mathbf{X} \left(\mathbf{a}^\top \mathbf{X}\right)^\top$$
$$= \underset{\mathbf{a}}{\arg\max}\ \mathbf{a}^\top \mathbf{\Sigma}^\mathbf{x} \mathbf{a}, \tag{1}$$

where $\mathbf{X} \in \mathbb{R}^{T \times N}$ is the data matrix containing $N$, $T$-dimensional samples, $\mathbf{\Sigma}^\mathbf{x} = \mathbf{X}\mathbf{X}^\top$ is the *scatter matrix*, and $\mathbf{a}$ is constrained to unit norm, i.e., $\|\mathbf{a}\|_2^2 = \mathbf{a}^\top \mathbf{a} = 1$. Differentiating the corresponding Lagrangian $\mathcal{L}(\mathbf{a}, \lambda) = \mathbf{a}^\top \mathbf{\Sigma}^\mathbf{x} \mathbf{a} - \lambda \mathbf{a}^\top \mathbf{a}$ with respect to $\mathbf{a}$ and equating to zero yields the following eigenvalue problem [8]:

$$\mathbf{\Sigma}^\mathbf{x} \mathbf{a} = \lambda \mathbf{a}, \tag{2}$$

where $\lambda$ is the largest eigenvalue of $\mathbf{\Sigma}^\mathbf{x}$ and $\mathbf{a} = \mathbf{a}^\star$ is the corresponding eigenvector. Thus, the maximal variance $\lambda$ is obtained along the principal direction $\mathbf{a}^\star$. The full eigenvalue problem can be written as:

$$\mathbf{\Sigma}^\mathbf{x} \mathbf{A} = \mathbf{A}\mathbf{\Lambda}, \tag{3}$$

where $\mathbf{A}$ is an orthonormal matrix and $\mathbf{\Lambda}$ is a diagonal matrix with eigenvalues ordered from largest to smallest, each corresponding with a column of $\mathbf{A}$. The columns are orthogonal because variance along the $i$-th direction $\mathbf{a}_i^\star$ is maximized in the space orthogonal to all preceding $\left[\mathbf{a}_1^\star \cdots \mathbf{a}_{i-1}^\star\right]$. Under orthogonality, we can write $\mathbf{A}^{-1} = \mathbf{A}^\top$.

In the case of real-valued data, the scatter matrix eigenvalue problem above is intimately related to the singular value decomposition (SVD) [9] of $\mathbf{X}$. Specifically, they share the same basis $\mathbf{A}$:

$$\mathbf{X} = \mathbf{A}\mathbf{P}\mathbf{Y}^\top, \tag{4}$$

where $\mathbf{A}$ is also known as the "left" singular vectors of $\mathbf{X}$, $\mathbf{P} = \mathbf{\Lambda}^{\frac{1}{2}}$ are the singular values, and $\mathbf{Y}$ is an orthonormal matrix containing the "right" singular vectors of $\mathbf{X}$. PCA then consists of aligning the principal axes of the data with the canonical basis by transforming (i.e., rotating) the data with $\mathbf{A}^\top \mathbf{X} = \mathbf{P}\mathbf{Y}^\top$, which preserves the data variance. Further normalizing with $\mathbf{P}^{-1}\mathbf{A}^\top \mathbf{X} = \mathbf{Y}^\top$ sphericizes the variances, yielding $\mathbf{\Sigma}^\mathbf{y} = \mathbf{I}$, which is referred to as "whitening" because the variances are equal along $\mathbf{Y}$.

Equivalently, solving the eigenvalue problem on $\mathbf{X}^\top \mathbf{X}$ yields $\mathbf{Y}$ directly, and $\mathbf{A} = \mathbf{X}\mathbf{Y}\mathbf{P}^{-1}$. We often solve the eigenvalue problem on the matrix ($\mathbf{\Sigma}^\mathbf{x}$ or $\mathbf{X}^\top \mathbf{X}$) with the shortest dimension for best accuracy, limiting the decomposition to the top $k$ eigenvalues for computational economy [10]. Group PCA (gPCA) is merely solving the eigenvalue problem on $M$ concatenated datasets, i.e., $\mathbf{X} = \left[\mathbf{X}_1^\top \ldots \mathbf{X}_M^\top\right]^\top$.

### B. Considerations for Analysis of Group FMRI Data

*1) Pre-processing:* PCA is used to reduce the dimensionality of subject data while maintaining most of its variability and achieve a condensed representation. Group PCA extends single-subject PCA to combine spatial maps across individuals, capturing prominent features from group data (Fig. 1).

Since PCA is entirely driven by variance, it follows that scaling differences between subjects are a key factor to determine their individual contributions and influence at the group level. The biasing influence of scaling effects in the case of site- or scanner-induced image intensity variability has been studied extensively in the field of data harmonization, leading to pre-processing tools such as ComBat [11] and traveling-subject based measurement bias correction [12]. But even when site- and scanner-induced intensity scaling effects are absent, the extent to which a group of subjects influences the results can raise questions about fairness and representation. This is especially concerning when studying minority and hard-to-scan populations.

Simple sum-of-squares normalization [13] and subject-level PCA with whitening [10] are two commonly employed pre-processing approaches to balance the individual contribution of subjects to the gPCA result. The main appeal of setting the total sum-of-squares (aka, the Frobenius norm) of each subject-specific dataset to 1 is that it ensures the total "variance" contribution from each subject is exactly the same. The net effect of this approach is that the relative "weigh" of subject-specific principal components is preserved at the group level. On the other hand, "whitening" in subject-level PCA enforces equal weight (i.e., unit variance) for all subject-specific principal components. Note that the total "variance" contribution per subject is also the same with this approach. The key difference is that the relative contribution of subject-level principal components is *not* preserved.

*2) Memory, Equivalences, and Methods:* Historically, the amount of RAM in desktops and workstations was insufficient for concatenating all subject datasets into a single matrix for singular value decomposition, which inspired a number of approaches for memory-efficient group PCA [14], culminating into low-RAM, highly accurate iterated approaches like SMIG [15] and MPOWIT [10]. Previously, two- and even three-step PCA approaches were common, proposing to use PCA reduction successively on subjects and subgroups as a means to reduce the memory footprint with "intermediate" subgroup representations before the final group estimation. Soon it was noted [15] that combining representations from subgroup compression with intermediate PCA steps did *not* yield the same group-level estimates as when all data were concatenated into a large, single step gPCA.

Incremental approaches like MIGP [15] and STP [10] emerged to address this issue, observing that it was necessary to carry the subgroup-level singular values into the final gPCA step (i.e., no whitening to force variances to 1) to obtain the correct estimates. For this reason, concerns have also been raised about the use of whitening in subject-level PCA, but no study so far has addressed this question pragmatically. Meanwhile, evidence for the utility of subject-level PCA with whitening as a preprocessing tool remains empirical.

Related back-reconstruction strategies to recover subject-level components from group-level estimates have since been studied but in the context of group independent component analysis (ICA) [16], which we do not consider in this work.

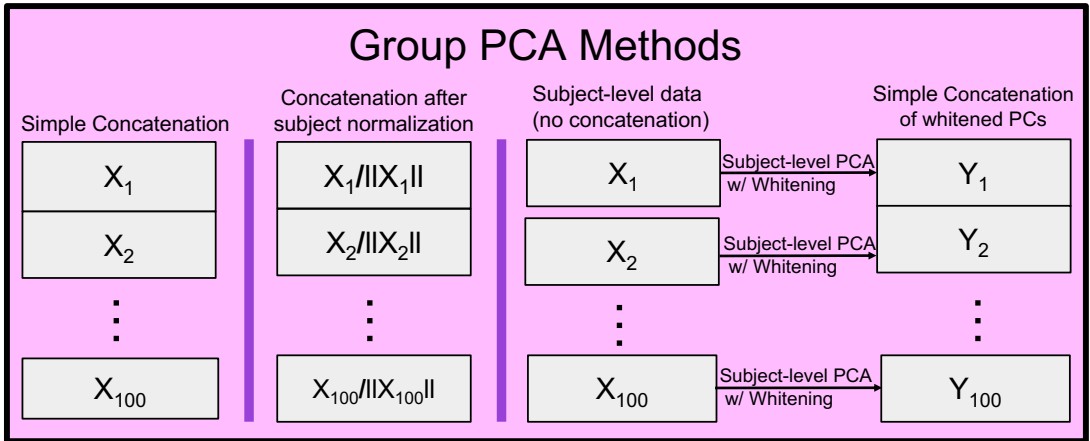

Fig. 1. **The three group PCA implementations evaluated in this work**. (a) Simple concatenation of subject data, (b) concatenation after subject sum-of-squares normalization, and (c) concatenation after subject-level PCA with whitening. For the latter, subject-level proportion of variance retention thresholds are set at 1, 0.99, 0.9, and 0.65, progressively reducing the subject-level information carried into group PCA.

## C. Our Contributions

The goal of our experiments is to determine which gPCA approach can recover subject-level component variability closest to the ground-truth group mean performance. Unlike previous studies, we follow a set of principled simulation guidelines for neuroimaging as outlined in [17], which helps minimize confounds among various configuration settings. First, we provide a generative model of subject-level PCA where the principal components of each subject are linked across subjects. This link is established by specifying cross-subject correlation matrices for each principal component. Using the proposed generative model, our approach allows full control of the statistical properties of every simulated principal component, including 1) their variance, 2) their similarity levels across subjects, and 3) the variance profile across subjects. In the interest of minimizing the chances of uncontrolled confounds, we restrict the use of additive noise to the subspace orthogonal to the signal subspace, but we do not impose theoretical limits on the variance assigned to noise.

The proposed controlled experimental setting enables a complete characterization of how latent dissimilarities among subjects and variance-driven biases affect the recovered subject-level variability, compared to what could be recovered using the ground-truth group mean. With that, we hope to determine preliminary recommendations for preprocessing.

## III. METHODS

### A. A generative model for multi-subject PCA

Based on the SVD decomposition in (4) and the expectation that latent sources are linked/similar across individuals, we propose the following generative model for multi-subject PCA:

$$\mathbf{X}_m = \mathbf{A}_m \mathbf{P}_m \mathbf{Y}_m^\top, \tag{5}$$

where $\mathbf{Y}_m = \left[\mathbf{y}_m^1 \cdots \mathbf{y}_m^i \cdots \mathbf{y}_m^C\right]$ contains the $C$ latent sources $\mathbf{y}_m^i$ of subject $m$. Each latent source is sampled from an $M$-dimensional unit variance multivariate Gaussian

distribution, with non-zero correlation structure (different per source $i$) describing the strength of the link/similarity among $M$ subjects. The $m$-th element of the $i$-th source is assigned to the $m$-th subject as $y_m^i$ for each of the $V$, $M$-dimensional samples generated per source, such that $\mathbf{Y}_m \in \mathbb{R}^{V \times C}$. $\mathbf{P}_m$ is a diagonal matrix where each diagonal element $P_m^i$ is a real, positive value that describes the standard deviation of source $i$ in subject $m$ (ordering the sources by their standard deviation is considered irrelevant, as long as their index $i$ is consistent across subjects). Finally, the random orthonormal matrix $\mathbf{A}_m = \left[\mathbf{a}_m^1 \cdots \mathbf{a}_m^i \cdots \mathbf{a}_m^C\right]$, $\mathbf{A}_m \in \mathbb{R}^{T \times C}$, defines the basis along which the sources are distributed, unique for each subject $m$.

Here, we set $C_n$ out of $C$ latent sources as noise by sampling them from i.i.d. $M$-dimensional white Gaussian noise (uncorrelated across subjects). The remaining $C_s = C - C_n$ sources are considered as signal. In this type of simple, separable noise structure, the noise is completely orthogonal to the signal, which serves as a fair, non-confounded setting for this initial investigation. Fig. 2 provides a depiction of the model in (5). The experiments were repeated with eleven different random sampling seeds, each yielding a new dataset.

### B. Experimental Settings

We generate data according to the proposed generative model above. First, we simulate ground-truth spatial maps with $V = 22,341$ voxels, $T = C = 50$ sources per subject, and $M = 100$ subjects. Here, $C_s = 40$ components represent signal sources and $C_n = 10$ components represent noise sources. The signal source similarity among subjects is systematically reduced from high in the first source ($i = 1$) to very low in the fortieth ($i = 40$).

We manipulate three key parameters to evaluate the capabilities of three gPCA methods in terms of retained subject-specific variance. First, the signal source proportion ($p_s$) represents the ratio of signal to total variance in the data. This proportion models real-world scenarios where the ratios

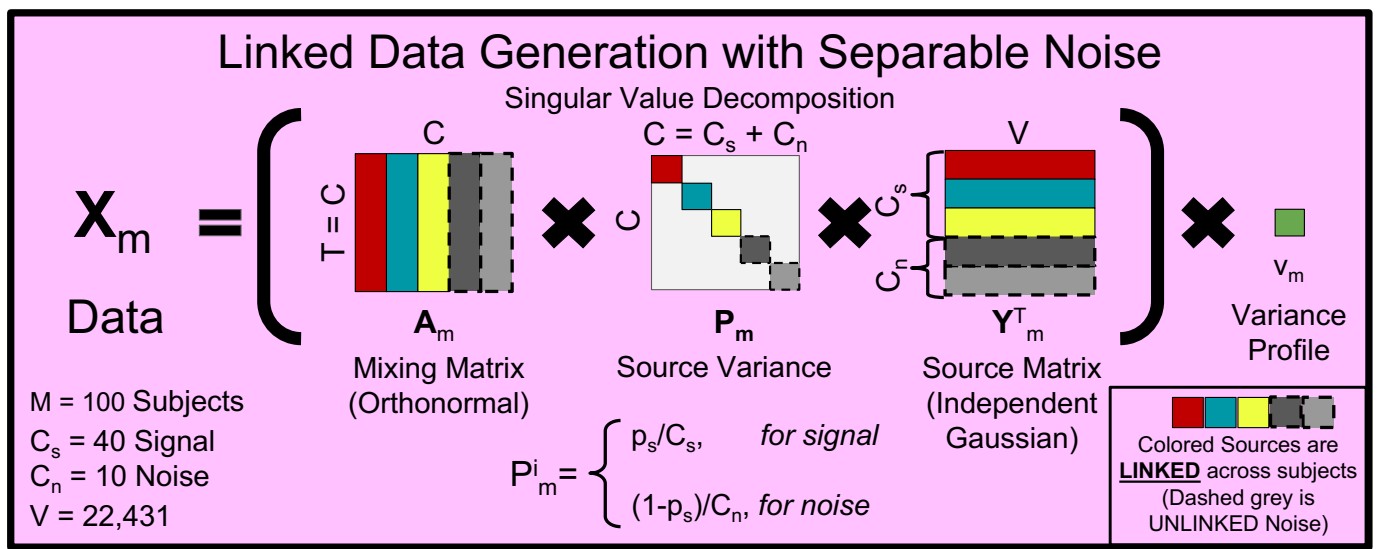

Fig. 2. **The data generation strategy using Singular Value Decomposition (SVD) for separable noise.** Ground-truth spatial maps are simulated with $V = 22,341$ voxels, $C_s = 40$ signal sources linked across $M = 100$ subjects (i.e., 40 independent 100-dimensional multivariate Gaussian distributions) at varying cross-subject similarity levels, and $C_n = 10$ independent noise sources per subject (i.e., 10 independent 1-dimensional Gaussian distributions per subject). The total number of sources is $C = C_s + C_n$. Signal sources are scaled equally (same variance) according to the source signal proportion $p_s$. Likewise for noise sources, but $1 - p_s$. When $p_s = 0.9$, the data is signal-dominant, while $p_s = 0.5$ balances signal and noise equally, and $p_s = 0.1$ makes noise dominant. $P_m^i$ determines the relative influence of signal and noise sources. The mixing matrix $\mathbf{A}_m$ of each subject is orthonormal and square ($T = C$), so it does not alter the total variance in the data. To generate the mixed data $\mathbf{X}_m$ of each subject, the mixing matrix multiplies the source matrix $\mathbf{Y}_m^\top$ after it is scaled by the source variance matrix $\mathbf{P}_m$, i.e., $\mathbf{X}_m = \mathbf{A}_m \mathbf{P}_m \mathbf{Y}_m^\top$. The mixed data of each subject is then scaled by a single number, $v_m$, to impose one of three variance profiles: 'Edge', 'Mid', or 'Rand'.

of signal and noise are unknown. The value defined for $p_s$ is divided uniformly across sources of each kind such that:

$$P_m^i = \frac{p_s}{C_s} \quad \text{and} \quad P_m^i = \frac{1 - p_s}{C_n}. \tag{6}$$

By setting the *noise* source proportion to $1 - p_s$ we ensure that the total variance of each subject is 1.

The second parameter, the variance profile $v_m$, changes the total variance of each subject and, thus, determines how the contributions of a subject are weighted at the group level, yielding:

$$P_m^i = v_m \frac{p_s}{C_s} \quad \text{and} \quad P_m^i = v_m \frac{1 - p_s}{C_n}. \tag{7}$$

The "Edge" profile emphasizes dissimilar subjects by increasing their $v_m$ relative to other subjects to counterbalance the similarity (i.e., the correlation) among subjects built into the signal sources. The "Mid" profile overemphasizes similar subjects by increasing their $v_m$, down-weighting dissimilar ones even further. The "Rand" profile randomly emphasizes subjects, mimicking realistic, unpredictable data distributions. The total variance across all subjects is set as $\sum_{m=1}^{M} v_m = M$.

The third parameter, the threshold for proportion of variance retained post subject-level PCA with whitening, sets the number of subject-specific principal components carried over to the group analysis, enabling identification of optimal component selection strategies. This pragmatic evaluation can inform improvements in gPCA methodology.

### C. Non-separable noise

Here we describe experiments to assess the effect of noise leaking into the signal subspace. We simulated $C_s = 40$ signal sources $\mathbf{Y}_m^{signal}$ linked across subjects in the same way as the original experiment. We also simulated $C_n = 40$ i.i.d. noise sources per subject $\mathbf{Y}_m^{noise}$ in the same way as before. The key difference is that now we simulated only 40 random orthonormal directions per subject to define the basis $\mathbf{A}_m$ and, after we adjust the variances of signal and noise sources, we use the same $\mathbf{A}_m$ matrix to mix the signals and the sources, i.e., $\mathbf{X}_m = \mathbf{A}_m \mathbf{P}_m \mathbf{Y}_m^{signal} + \mathbf{A}_m \mathbf{P}_m \mathbf{Y}_m^{noise}$.

### D. Evaluating performance

The formulation in (5) admits a source-wise representation:

$$\mathbf{X}_m = \sum_{i=1}^{C} \mathbf{X}_m^i = \sum_{i=1}^{C} \mathbf{a}_m^i P_m^i \left(\mathbf{y}_m^i\right)^T, \tag{8}$$

which computes the $i$-th source portion of the data matrix for a single subject. Using the aggregate source estimate $\mathbf{Y}_{gpca}$ from each of the group PCA methods considered here, we try to recover the variability of each source and each subject using linear regression:

$$\hat{\mathbf{X}}_m^i = \mathbf{X}_m^i \mathbf{Y}_{gpca} \left(\mathbf{Y}_{gpca}^T \mathbf{Y}_{gpca}\right)^{-1} \mathbf{Y}_{gpca}^T. \tag{9}$$

The sum of squared error between $\mathbf{X}_m^i$ and $\hat{\mathbf{X}}_m^i$, relative to the total sum of squares of $\mathbf{X}_m^i$, yields the normalized sum of

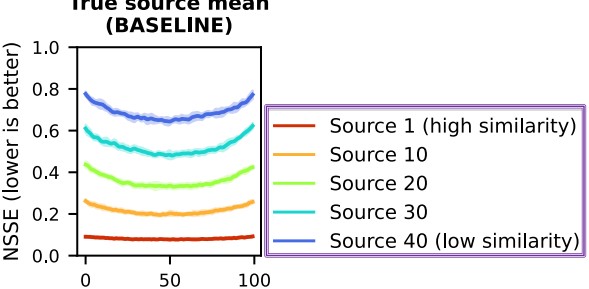

Fig. 3. **The ground-truth mean performance**. The ground-truth mean is defined as the mean over subjects of their standardized, unit-variance sources. The y-axis represents the NSSE obtained by utilizing the ground-truth mean to approximate source-specific variability in each subject. The ground-truth mean captures very well the variability for sources of high similarity across subjects (red), but not so for sources that are more dissimilar across subjects (blue). For clarity, only results for sources 1, 10, 20, 30, and 40 are shown. In addition, the approximation error is more accentuated for subjects at the "edges" of the plot, i.e., the most dissimilar from the subjects in the middle (the subjects in the middle are always the most similar by design).

squares error (NSSE) for a single subject and source, which we used as a key performance metric in this work:

$$\text{NSSE} = \frac{\sum_{t=1}^{T} \sum_{v=1}^{V} \left( \hat{\mathbf{X}}_m^i - \mathbf{X}_m^i \right)^2}{\sum_{t=1}^{T} \sum_{v=1}^{V} \left( \mathbf{X}_m^i \right)^2}. \tag{10}$$

NSSE is a ratio of error power with respect signal power specific to each source and each subject, describing how much variance is recovered by the aggregate measure for a specific source and subject. This way we hope to determine what specific variances are lost by each method and condition tested. Our baseline for comparing gPCA methods is the NSSE for the ground-truth mean of each source:

$$\mathbf{Y}_{\mu_{GT}} = \frac{1}{M} \sum_{m=1}^{M} \mathbf{Y}_m. \tag{11}$$

A small value indicates little error, and a large value a large error. To determine the performance of gPCA methods relative to the baseline $\mathbf{Y}_{\mu_{GT}}$ we define the *relative* NSSE as the ratio:

$$\text{NSSE}_{rel} = \frac{\text{NSSE}_{gpca}}{\text{NSSE}_{\mu_{GT}}}. \tag{12}$$

This allows us to determine source- and subject-specific variability lost by a gPCA method relative to the corresponding ground-truth source mean performance. An $\text{NSSE}_{rel} = 1$ indicates the gPCA method produces the same result as the baseline. A value larger than 1 indicates the gPCA method produces a degraded result (more error), and a value smaller than 1 indicates the gPCA method produces a better result. By comparing variance retention between gPCA approaches and the ground-truth mean sources, we quantify subject-level variability loss and assess group-level source accuracy.

## IV. RESULTS

In Fig. 3 we show the subject-level variance lost by the ground-truth source mean. Sources with high similarity across

subjects are represented by red and sources with low similarity are represented by blue. The experiments were repeated with eleven different random sampling seeds, each yielding a new dataset. The band-plots capture the fifteenth to eighty-fifth percentiles of variability among the eleven seeds.

We observed that the NSSE was smaller for sources of high similarity across subjects, indicating good recovery of subject-specific variability. On the other hand, there is a clear trend of higher NSSE for sources with lower similarity across subjects. This indicates that even the ground-truth group mean can lead to biased inferences about subjects that deviate from the population average, especially for subjects at the "edges" of the plot (i.e., those with limited inter-subject similarity).

In Fig. 4, we show the effects of signal source proportion ($p_s$) on simple concatenation gPCA, subject sum-of-squares normalization gPCA, and concatenation gPCA following subject-level PCA with whitening. Sources with high similarity across subjects are represented by red and sources with low similarity are represented by blue. The band-plots capture the fifteenth to eighty-fifth percentiles of variability among the eleven seeds. The stars on the panels represent performance closely matching the ground-truth mean.

Our first observation pertaining to proportion of subject-level variance lost by gPCA relative to ground-truth mean (Fig. 4) is that most gPCA methods and specifications in situations of low to medium noise consistently match the ground truth mean performance very well. For example, in the signal-dominant $p_s = 0.9$ condition, most gPCA methods and specifications perform comparable to the ground-truth mean. Only gPCA post subject PCA whitening at 0.65 data reduction loses significant information for subjects with high similarity. In the balanced signal to noise condition ($p_s = 0.5$), performance matches the signal dominant condition except for post subject PCA whitening at or below 0.9 data reduction, for which more information from subjects with high similarity is lost. Lastly, in the noise-dominant $p_s = 0.1$ condition, most gPCA methods and specifications lose significant information from subjects. Only gPCA post subject PCA whitening with no data reduction completely matches the ground truth mean performance in extreme levels of noise.

This result provides an argument for subject-level PCA whitening due to its ability to perform on par with the ground-truth mean, retaining comparable subject-level variability even in scenarios of extreme noise. While this method seems very sensitive to certain variance retention thresholds, we perceive this simply as an indication that subject-level reduction after whitening is unnecessary and not recommended prior to group PCA, based on the provided evidence.

Our second observation is that in extreme noise the other gPCA approaches capture variability from subjects with low similarity better than subjects with high similarity (inverted U-shape). This pattern replicated in non-separable noise (Fig. 5).

Lastly, we observe that the 'Edge' variance profile for gPCA simple concatenation had the least desirable effect on the sources. In the 'Mid' profile, it captured subject-level variability better than the ground truth mean for some

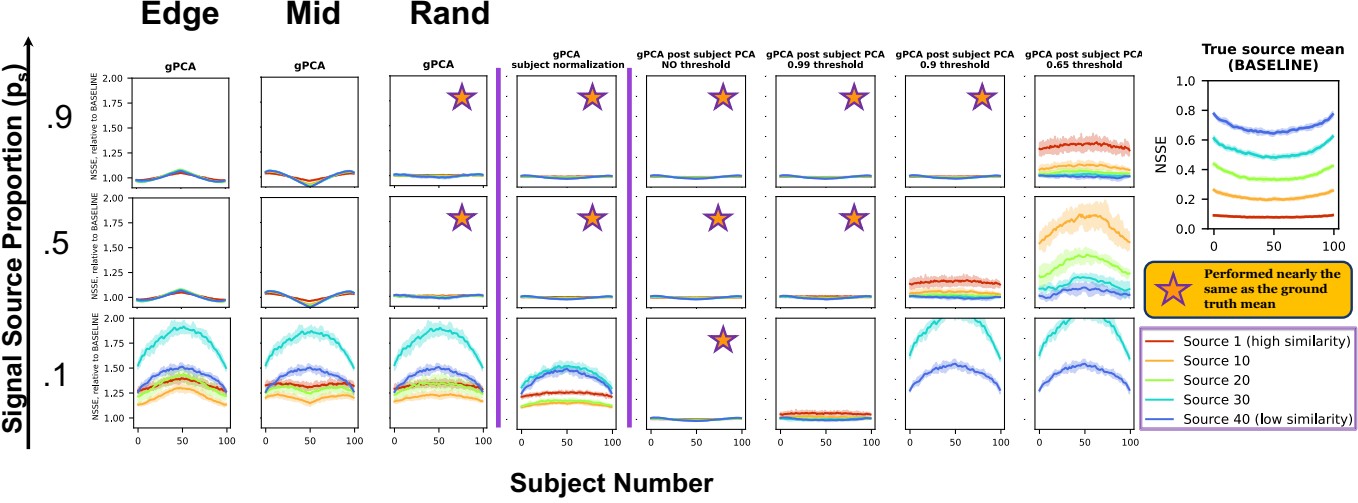

Fig. 4. **The performance of each group PCA approach relative to the ground-truth mean baseline performance for each subject, at different noise levels and variance profiles**. The y-axis in each subpanel represents the relative Normalized Sum of Squares Error with respect to the baseline for a single subject and source. The x-axis represents each of the $M = 100$ subjects. Subpanels are organized by column and row. Each column represents a gPCA method with varying specifications while each row represents the simulated signal source proportion ($p_s$), i.e, the proportion of data variance assigned to signal sources. The first three subpanel columns correspond to simple concatenation (standard gPCA) with the simulated variance profiles set to either 'Edge', 'Mid', or 'Rand', respectively. The fourth column is for gPCA with subject-level sum-of-squares normalization. The last four columns correspond to gPCA following subject-level PCA with whitening and either no data reduction, or data reduction at the 0.99, 0.9, and .65 levels, respectively. Subpanels with a star closely replicate the ground-truth mean performance for all subjects and all sources. Each line is the average over the 11 random seeds. For clarity, only results for sources 1, 10, 20, 30, and 40 are displayed; the simulated level of similarity between subjects for each source decreases with the source number by construction. Note: only simple concatenation gPCA is affected by variance profile. Bands capture the core 70% of performance variation across seeds.

subjects, and in the 'Rand' profile it consistently matched the performance of other gPCA methods, except in extreme noise. A key observation here is that simple concatenation gPCA's performance is indeed sensitive to variance profile, which is undesirable in the sense that it is affected by arbitrary scaling effects at the subject level and gives preferential treatment to certain subjects.

In Fig. 5, the preliminary result suggests that all gPCA methods degrade in performance and fail to match the ground-truth source mean even at medium signal proportion level ($p_s = 0.5$). There is no obvious prevailing method like in the separable noise case, and failure levels are highly similar across methods. As in the separable noise case, simple concatenation gPCA is highly sensitive to different variance profiles, whereas the other methods are not.

## V. PRELIMINARY RECOMMENDATIONS

Given the sensitivity to variance profile changes observed for conventional gPCA, our first recommendation is utilizing subject-level normalization to eliminate variance profile effects in simple concatenation gPCA. This is a very cheap solution (simply normalize the sum-of-squares for each subject) that addresses a largely overlooked issue with likely detrimental consequences since the variance profile of a dataset is, in principle, not a tunable experimental parameter, just a property of the dataset. In other words, just utilize gPCA with subject-

level normalization. It has very low computational overhead compared to simple concatenation gPCA and delivers significant reduction in variance profile effects.

Our second recommendation is doing little to no data reduction for gPCA with subject-level whitening, especially if the goal is to match the ground truth mean as best as possible. While closer to the ground-truth performance, this approach effectively requires about twice the computational effort.

## VI. CONCLUSION

Subject-level PCA with whitening and no data reduction perfectly matches ground-truth mean performance under separable noise, even with extreme noise levels, effectively denoising data in this simple setting. This result provides more insight into the functionality of subject-level PCA with whitening, and an argument towards using whitening as long as there is no data reduction. It is important to note that any reduction after whitening appears to significantly affect performance relative to other methods and in proportion to the level of noise in the data. Subject sum of squares normalization performs adequately throughout our experiments, and simple concatenation performs least desirably in terms of approximating the ground-truth mean performance. It should be emphasized that no gPCA method uniformly outperforms the ground-truth source mean, and the ground-truth source mean itself can lead to biased inferences about subjects that

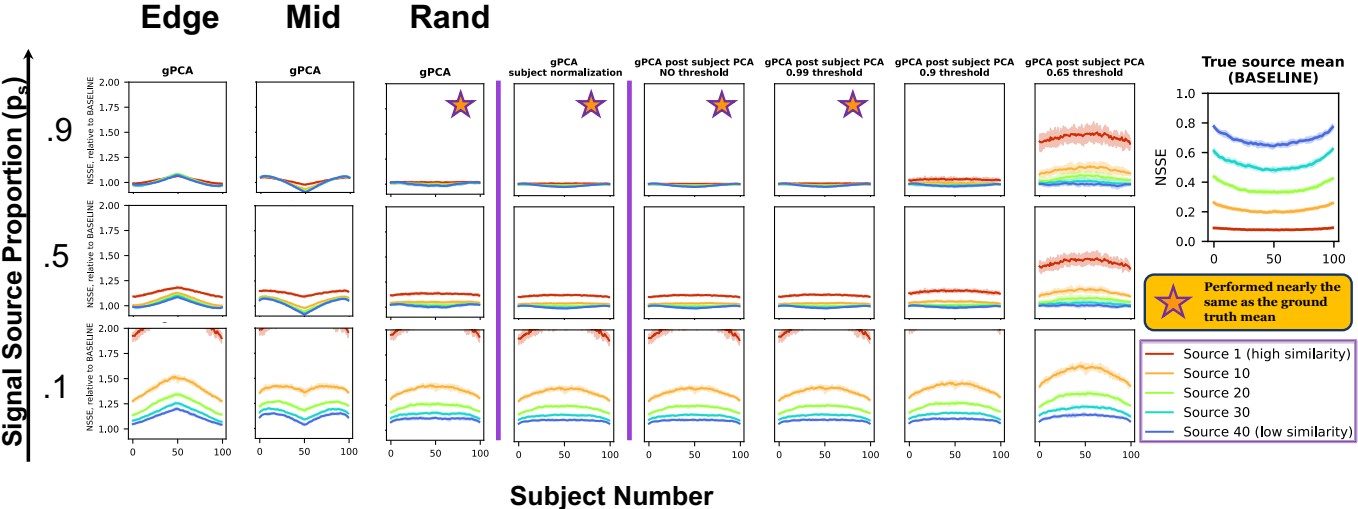

Fig. 5. **The performance of each group PCA approach relative to the ground-truth mean baseline performance for each subject, at different levels of non-separable noise and variance profiles**. The axes and subpanels are organized in the same way as Fig. 4. Subpanels with a star closely replicate the ground-truth mean performance for all subjects and all sources. For clarity, only results for sources 1, 10, 20, 30, and 40 are displayed. Note: only simple concatenation gPCA is affected by variance profile ('Edge', 'Mid', or 'Rand').

deviate from the population average. Although the current iteration of our work does not tackle the full, challenging nature of real fMRI data, where no ground-truth is available, it does offer a novel, highly objective framework for testing key factors that are highly relevant to the gPCA approaches we investigated and, even at this initial stage, it reveals how certain data characteristics influence performance for different implementations.

Future work will explore more challenging noise scenarios such as correlated noise and evaluate the performance of gPCA methodology on real data. A full evaluation of challenging realistic noise structures will provide insight on the capabilities and limitations of gPCA to elicit new guidelines for neuroimaging analyses.

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
