# OpenReview forum: "Robust Group PCA for Separable Noise: An Argument for Subject-Level PCA"
_IEEE.org/EMBS/BHI/2025/Conference — BHI 2025_

### Official Review · Reviewer_Dzik · 2025-07-17
**A comparison of group PCA methods demonstrates the ability of concatenation with subject-level PCA whitening to near-ground truth**

**Confidence:** 4
**Clarity Of Writing:** great
**Clinical Significance:** fair
**Methodological Novelty:** fair
**Overall Rating:** 7
**Final Rating:** 7

**Experiments And Results:**

good

**Questions For The Authors:**

The authors mention that, at times, the ground truth mean can lead to bias for specific subjects that tend away from the population mean. Would it perhaps be better to therefore compare on a more granular, subject-to-subject basis?

**Strengths:**

The testing assessing the various gPCA methods and resistance to data reduction is fairly robust. This found that there needs to be further testing and tuning to better understand when one may be a more optimal choice when challenged with different datasets. The methodology is also fairly well-explained.

**Summary Of The Paper:**

In the datasets with low starting variability, all the gPCA methods performed similarly to ground truth. With high noise, the gPCA after subject-level whitening performed closely to the ground truth mean. However, the caveat is with data reduction, variability gets lost with this combination.

**Weaknesses:**

As the paper outlines, future work will be needed to better understand which gPCA methods (or how they should be altered) to be less susceptible to both noise and data reduction.

---

### Official Review · Reviewer_SYJs · 2025-07-18
**Important problem motivation, with lack of real world data**

**Confidence:** 4
**Clarity Of Writing:** good
**Clinical Significance:** good
**Methodological Novelty:** fair
**Overall Rating:** 6
**Final Rating:** 7

**Experiments And Results:**

good

**Questions For The Authors:**

N/A

**Strengths:**

The paper addresses a critical issue in large-scale neuroimaging

The authors propose a principled simulation framework to rigorously test various gPCA approaches using a well-controlled generative model

The paper highlights equity and fairness implications, such as the potential exclusion of minority subjects in aggregated models

**Summary Of The Paper:**

This study investigates how different implementations of group Principal Component Analysis (gPCA) affect the accuracy and robustness of group-level fMRI representations, particularly in preserving subject-specific information

**Weaknesses:**

The study is confined to separable noise scenarios; real-world fMRI data often include complex, non-separable noise patterns, limiting immediate applicability.

All experiments are conducted on simulated datasets. While simulations are rigorous, validation on empirical fMRI datasets would strengthen the practical impact.

 Although whitening performs best without data reduction, its sensitivity to hyperparameters is a limitation

---

### Official Review · Reviewer_ZveA · 2025-07-18
**A thorough evaluation of the paper, highlighting its innovative approach to group PCA in fMRI data analysis, with suggestions for improving clarity, experimental validation, and generalizability.**

**Confidence:** 3
**Clarity Of Writing:** great
**Clinical Significance:** great
**Methodological Novelty:** great
**Overall Rating:** 7

**Experiments And Results:**

great

**Questions For The Authors:**

In the proposed generative model, restricted additive noised to the subspace orthogonal to the signal subspace., how might including noise in the subspace affect the results?

**Strengths:**

The proposed generative model for multi-subject PCA is a significant contribution, allowing precise control over statistical properties such as variance, cross-subject similarity, and noise. This enables a robust framework for evaluating gPCA methods under controlled conditions, which is a valuable tool for future research.
Preserving subject-specific variability is a critical issue in fMRI research, where group-level analyses often obscure individual differences, particularly in diverse or minority populations. This is key factor for diversity of fMRI datasets.

**Summary Of The Paper:**

This paper investigates the use of group Principal Component Analysis (gPCA) to aggregate functional magnetic resonance imaging (fMRI) datasets, with a focus on preserving subject-specific variability. The traditional gPCA methods, such as simple concatenation or sum-of-squares normalization, may obscure individual differences, potentially biasing group-level estimates and limiting the applicability of findings, especially in diverse populations.
A novel approach is proposed that involves subject-level PCA combined with whitening, along with a generative model to simulate multi-subject fMRI data featuring controlled statistical properties. There are 3 gPCA implementations:
•	Simple concatenation,
•	Sum-of-squares normalization, and
•	Subject-level PCA with whitening

These are evaluated using a simulated dataset of 100 subjects, 40 signal sources, and varying levels of cross-subject similarity. The impact of these methods on capturing subject-level variability relative to a ground-truth mean is assessed using Normalized Sum of Squared Errors (NSSE). The results indicate that subject-level PCA with whitening better preserves individual variability, particularly among dissimilar subjects, supporting preprocessing strategies. These will account for latent dissimilarities and variance-driven biases.

**Weaknesses:**

This study relies on simulated data, which offers control over specific variables but may not fully reflect the complexity and variability of real-world fMRI datasets.

---

### Official Review · Reviewer_ryMg · 2025-07-18
**Simulation only evaluation limits real-world applicability**

**Confidence:** 3
**Clarity Of Writing:** good
**Clinical Significance:** great
**Methodological Novelty:** good
**Overall Rating:** 7

**Experiments And Results:**

great

**Questions For The Authors:**

1. How do results change with correlated noise typical in fMRI?
2. What guidelines exist for threshold selection in real datasets?
3. Why no validation on real fMRI data?

**Strengths:**

1. The authors address preserving subject-specific information in group analyses, critical for minority populations.
2. The authors use rigorous simulations with controlled statistical properties.
3. The authors evaluate methods in widely-used toolboxes (FSL, GIFT).

**Summary Of The Paper:**

The authors compare three group PCA methods for combining fMRI data across subjects. Using simulations, they find subject-level PCA whitening works best but is sensitive to parameter choices.

**Weaknesses:**

1. The authors limit analysis to separable noise, unrealistic for real fMRI data.
2. The authors provide no real data validation.
3. The authors give insufficient guidance for threshold selection.
4. The authors omit computational cost analysis.